# Role of Probiotics in Preventing Carbapenem-Resistant *Enterobacteriaceae* Colonization in the Intensive Care Unit: Risk Factors and Microbiome Analysis Study

**DOI:** 10.3390/microorganisms11122970

**Published:** 2023-12-12

**Authors:** Jung-Hwan Lee, Jongbeom Shin, Soo-Hyun Park, Boram Cha, Ji-Taek Hong, Don-Haeng Lee, Kye Sook Kwon

**Affiliations:** 1Division of Gastroenterology, Department of Internal Medicine, Inha University Hospital, Inha University School of Medicine, Incheon 22332, Republic of Koreashinjongv@naver.com (J.S.); hahahjt2@naver.com (J.-T.H.); 2Department of Hospital Medicine, Inha University Hospital, Inha University School of Medicine, Incheon 22332, Republic of Korea; 3Department of Neurology, Kangdong Sacred Heart Hospital, Hallym University College of Medicine, Seoul 05355, Republic of Korea; g2skhome@hanmail.net

**Keywords:** probiotics, risk factor, carbapenem-resistant *Enterobacteriaceae*, prevention

## Abstract

Older patients with multiple comorbidities often necessitate prolonged hospital stays and antibiotic treatment in the intensive care unit (ICU), leading to a rise in multidrug-resistant organisms like carbapenem-resistant *Enterobacteriaceae* (CRE). This study examined risk factors for carbapenem-resistant Enterobacteriaceae colonization in the ICU and assessed probiotics’ preventive role. In this single-center, retrospective study, 9099 ICU patients were tested for stool CRE culture from March 2017 to April 2022. We excluded 136 patients with CRE colonization within one week post-admission and 26 who received probiotics before CRE colonization. Ultimately, 8937 CRE-negative patients were selected. Logistic analysis identified CRE colonization risk factors and evaluated probiotics’ influence, including *Saccharomyces boulardii* or *Lactobacillus rhamnosus*, used by 474 patients (5.3%) in the ICU. Compared with data on initial admission, 157 patients (1.7%) had newly discovered CRE colonization before discharge. In a multivariate analysis, coronavirus disease 2019, the ICU, tube feeding, antibiotics such as aminoglycoside, extended-spectrum penicillin, stool vancomycin-resistance *enterococci* colonization, and chronic kidney disease were significantly associated with de novo CRE infection. However, probiotic use was negatively correlated with CRE infection. Managing risk factors and administering probiotics in the ICU may help prevent CRE colonization; large randomized prospective studies are needed.

## 1. Introduction

Carbapenem-resistant *Enterobacteriaceae* (CRE) are antibiotic-resistant bacteria that are increasingly common in healthcare settings. According to the Centers for Disease Control and Prevention, the incidence of CRE infections in the USA has increased in recent years. The percentage of *Enterobacteriaceae* isolates resistant to carbapenems increased from 1.2% in 2001 to 4.2% in 2017 [1]. Furthermore, patients in the intensive care unit (ICU) have a higher risk of developing healthcare-associated infections owing to a variety of factors, such as underlying illnesses, use of antibiotics, and prolonged hospital stays. In Korea, the incidence of CRE colonization in ICUs varied, with a carriage rate upon admission ranging from 2.8% to 6.3% [2]. A systematic review indicates an approximate 16.5% risk of infection with CRE among patients colonized with CRE, accompanied by a high mortality rate associated with CRE infections [3]. Therefore, preventing or managing CRE in the ICU is significantly associated with patient safety. 

CRE colonization in feces is difficult to treat and can cause severe infections. Notably, fecal microbiota transplantation (FMT) was hypothesized to be a possible treatment option for CRE decolonization [4,5,6]. These previous studies, including our published FMT study, have shown promising results in using FMT for CRE decolonization, with others reporting a successful eradication of CRE from the gastrointestinal tract in a significant proportion of treated patients. However, FMT use for CRE decolonization remains an investigational treatment and has not yet been approved for routine clinical use. Additionally, FMT is a complex medical procedure that should only be performed by qualified healthcare professionals in appropriate clinical settings. 

A potential strategy to prevent the spread of CRE is using probiotics, which are live microorganisms that confer health benefits to a host when administered in sufficient quantities; specifically, they are believed to promote gut health by balancing the microbiota and preventing the overgrowth of harmful bacteria [7]. The use of probiotics to prevent CRE fecal colonization has been the subject of several studies; however, most have not demonstrated a significant effect [8]. Nonetheless, these studies had small sample sizes and lacked long-term follow-ups and microbiome analyses. Therefore, this study aimed to analyze the risk factors for CRE fecal colonization in the ICU and to investigate the effectiveness of probiotics in preventing colonization while exploring the potential mechanisms of the action of probiotics in preventing multi-drug-resistant organism (MDRO) colonization.

## 2. Materials and Methods

We performed a retrospective analysis of 12,355 patients initially admitted to the ICU of Inha University Hospital between March 2017 and April 2022 (Figure 1). Specifically, we included patients who had undergone a stool CRE culture within one week after admission and those who had undergone an initial stool CRE-negative test admission. Therefore, we selected 9099 patients who underwent a stool CRE culture during admission. A total of 136 and 26 patients with CRE colonization who underwent a stool CRE test 1 week after admission and a stool CRE test before probiotic administration, respectively, were excluded. Finally, 8937 patients with a negative stool CRE test within one week after admission were enrolled. This study was reviewed and approved by the ethics committee of Inha University Hospital (approval No. IUH 2023-07-023). Patients were not required to provide informed consent because this retrospective analysis used anonymous clinical and microbiologic data that were obtained after each patient agreed to treatment by written consent.

*Saccharomyces boulardii* and *Lactobacillus rhamnosus* were selected as probiotics because they have been shown to decolonize MDRO [7,9]. Furthermore, Inha University Hospital has operated a CRE screening program since 2017; therefore, all patients admitted to the ICU have been screened from admission to discharge every 15 days.

The change in stool CRE culture was defined as follows: (1) negative conversion, a switch from positive to two subsequent negative stool cultures until the last follow-up; (2) no change, the persistence of either a positive or negative test result; and (3) new onset CRE, a transition from negative to positive stool culture during admission. The following risk factors affecting new CRE colonization were considered: age, sex, ICU category, length of stay, tube feeding, probiotics administration and duration, proton pump inhibitor administration and duration, carbapenem administration and duration, antibiotics other than carbapenem, and comorbidity. 

Further, we selected patients with CRE colonization with or without probiotics for microbiome analysis using 16s RNA metagenomics sequencing. Notably, these samples were obtained from another MDRO study (KCT0004423). Moreover, fecal samples from 20 patients with CRE colonization were divided into the probiotic (12) and non-probiotic (8) groups. We conducted this study to investigate the effect of FMT on decolonizing CRE or vancomycin-resistant enterococci (VRE) [4]. From these samples, we selected the metagenomics data of 20 pre-FMT CRE stool samples. We analyzed α-diversity using two distinct methods: the species richness (CHAO1) and the diversity index (Shannon). The β-diversity analysis for both groups was executed using unweighted UniFrac and weighted UniFrac. Lastly, statistical analysis to determine the differences in β-diversities was conducted using the PERMANOVA method. This approach characterized the microbiome, including α- and β-diversities, and the relative abundance of specific taxonomic units associated with CRE, such as the *Enterobacteriaceae* family and the genera *Klebsiella, Escherichia, Citrobacter,* and *Enterobacter*.

### 2.1. Stool Bacterial Analysis for Metagenomics Analysis

A 16S rRNA amplicon sequencing with v3–4 primers was performed using a MiSeq sequencer on an Illumina platform (Macrogen Inc., Seoul, Republic of Korea) according to the manufacturer’s specifications. The original library and single long reads were obtained by assembling the paired-end reads created via sequencing in both directions, and FLASH (version 1.2.11) was used for this process [10]. Data containing sequence errors were removed for precise operational taxonomic unit (OTU) analysis. Reads containing ambiguous bases and chimeric sequences were removed because of the implied sequencing errors.

Among the assembled reads, those shorter than 400 or longer than 500 bp were excluded. Next, clustering was performed based on sequence similarity. The OTUs of the remaining reads were created using a cluster cutoff value of 97% in the CD-HIT-OTU program [10]. Moreover, data containing sequence errors were removed for precise OTU analysis and reads containing ambiguous bases and chimeric sequences were removed because of the implied sequencing errors.

Next, clustering was performed based on sequence similarity. The OTUs of the remaining reads were created using a cluster cutoff value of 97% in the CD-HIT-OTU program [11]. 

QIIME 2 and EzBioClould (CJ Bioscience Inc, Seoul, Republic of Korea; https://www.ezbiocloud.net/contents/16smtp (accessed on 23 August 2023) were used for OTU analysis and taxonomy [12,13]. The major sequence of each OTU was identified, and high-quality sequence reads were assigned to the “species group” at 97% sequence similarity using the PKSSU4.0 database. 

To confirm the diversity and evenness of the microbial community, the Shannon and Chao1 indices were calculated. β-diversity (diversity among samples within a group) was calculated based on the unweighted UniFrac distance and UniFrac. The genetic relationships among samples were visualized using principal coordinate analysis (PCoA).

### 2.2. Statistical Analysis

Using logistic analysis, we identified risk factors for CRE colonization and assessed the relationship between probiotics and CRE. A multivariate logistic regression with group assignment as a predictor variable and decolonization as the outcome variable was performed to calculate the odds ratios (OR) and 95% confidence intervals (CI). Fisher’s exact test was used to analyze categorical variables. Furthermore, differences in microbiota between the groups were analyzed using the Wilcoxon rank-sum test for two independent samples or the Wilcoxon signed-rank test for two related or matched samples. Data were analyzed using the R software (R Foundation, Vienna, Austria; https://www.r-project.org (accessed on 22 August 2023) or SPSS version 26 software (IBM Corp., Armonk, NY, USA). Finally, a permutational multivariate analysis of variance (PERMANOVA) was used to determine the significance of the PCoA plot results. Statistical significance was set at *p* < 0.05.

## 3. Results

The baseline characteristics of the enrolled patients are shown in Table 1. The patients were divided into two groups: CRE colonization and non-CRE colonization. A total of 157 patients (1.7%) showed new CRE colonization during admission. Age, ICU category, total length of stay, VRE colonization, *Clostridium difficile* infection, proton pump inhibitor usage and duration, carbapenem usage and duration, and tube feeding proportion significantly differed between the two groups. 

Among antibiotics, the frequencies of aminoglycosides, extended-spectrum penicillin, glycopeptides, second- and third-generation cephalosporins, quinolone, metronidazole, aztreonam, colistimate, tigecylcline, trimethoprim–sulfamethoxazole, azoles, and antibiotics duration were significantly different between the groups. Acute myocardial infarction and chronic kidney disease were also independent risk factors for new CRE colonization. Analyzing the genus composition of CRE colonization, the *Klebsiella* genus (120, 76.3%) constituted the largest portion of CRE.

### 3.1. Multivariable Analysis of Risk Factors Affecting New CRE Colonization

We determined the risk factors for new CRE colonization (Figure 2). In multivariate analysis, coronavirus disease 2019 (COVID-19) ICU (OR, 5.73; 95% CI, 2.11–15.55), tube feeding (OR, 2.16; 95% Cl: 1.44–3.25), extend spectrum penicillin (OR, 1.73; 95% Cl: 1.13–2.65), stool VRE colonization (OR, 1.64; 95% Cl: 1.08–2.5), and chronic kidney disease (OR, 1.3; 95% Cl: 1.06–1.59) were positively correlated with new CRE colonization. Moreover, quinolone (OR, 0.68; 95% Cl: 0.47–0.99), acute myocardial infarction (OR, 0.33; 95% Cl: 0.12–0.93), and probiotics administration (OR, 0.47; 95% Cl: 0.24–0.9) significantly and negatively correlated with new CRE colonization.

### 3.2. Analysis of the Microbiome’s Response to Probiotics in Patients with CRE Colonization

We analyzed the family and genus composition of CRE-colonized stool according to probiotics administration (Appendix A). The abundance of the Enterobacteriaceae family, to which CRE belongs, was lower in the probiotics group (3.43%) than in the non-probiotics group (20.85%). In contrast to the CRE genus composition, the abundances of the Klebsiella genus (<1 vs. 6.15%), Escherichia (2.51 vs. 1.76%), and Citrobacter (<1 vs. 4.04%) were different between the probiotics group and the non-probiotics group.

All α-diversities demonstrated using the Shannon and CHAO1 methods showed no significant differences between the probiotic and non-probiotic groups (*p* = 0.108 and *p* = 0.31, respectively) (Figure 3a,b). The β-diversity shown using unweighted UniFrac and weighted UniFrac was marginally different between the two groups (*p* = 0.054 and *p* = 0.046, PERMANOVA, respectively) (Figure 3c,d).

Further, we analyzed the relative abundances of both groups, including the *Enterobacteriaceae* family (Figure 4a) and the *Klebsiella* genus (Figure 4b), which are the main taxa of CRE. The relative abundances of the *Enterobacteriaceae* family in the probiotics group were lower than those in the non-probiotics group (*p* = 0.076). Similarly, the relative abundance of the *Klebsiella* genus in the probiotics group was significantly lower than that in the non-probiotics group (*p* = 0.043). However, the relative abundances of the Escherichia, *Citrobacter*, and *Enterobacter* genera were not significantly different (*p* = 0.673, *p* = 0.316, and *p* = 0.190, respectively) (Appendix A). 

## 4. Discussion

To our knowledge, this is the first study to demonstrate that probiotics had a preventive effect on CRE colonization using microbiome analysis in ICU patients. After adjusting for other risk factors, COVID-19 ICU, tube feeding, antibiotics such as aminoglycoside, extended-spectrum penicillin, glycopeptide, chronic kidney disease, and length of stay were significantly associated with de novo CRE infection in multivariate analysis. However, probiotics were negatively correlated with new CRE colonization. Moreover, we analyzed the microbiome associated with CRE abundance. Although we cannot directly demonstrate a decrease in the amount of CRE, the relative abundance of the *Enterobateriaceae* family in patients who received probiotics showed a reduction in the number of these taxa. In particular, the abundance of the *Klebsiella* genus, the main genus in CRE, significantly decreased. Therefore, probiotic administration may prevent new CRE colonization. 

CRE colonization in the ICU is a significant concern due to the potential for subsequent infections and limited treatment options. Particularly, in previous studies, several risk factors contributed to the colonization of CRE in ICU patients, such as antibiotics exposure, hospitalization, ICU stay, indwelling medical devices, and transfer from other healthcare facilities [14,15], consistent with our results. Our study proved COVID-19 ICU to be the most significant risk factor for CRE colonization. Prolonged hospitalization, broad-spectrum antibiotics, immunosuppressed status, shared environments, and close proximity were known as CRE-acquisition risk factors [16,17]. Longer lengths of ICU stay, carbapenem treatment, corticosteroid treatment, and invasive mechanical ventilation were frequently applied to COVID-19 ICU patients, increasing their susceptibility to CRE acquisition [17]. Additionally, elderly patients admitted to the COVID ICU were likely to be initially undetectable CRE carriers from long-term care facilities in the Korean medical environment [18]. Infection prevention management in the COVID ICU was also challenging due to the wearing of protective clothing and the risk of COVID-19 infection. Therefore, we should be more careful about CRE colonization in COVID-19 patients in the ICU. Various well-established prevention strategies can be implemented, such as hand hygiene, contact precautions, device control, environmental cleaning, and the early detection of CRE carriers using active surveillance methods [2]. In our hospital, UV light decontamination was routinely performed in the COVID-19 ICU to prevent the spread of CRE [19]. Additionally, stool VRE colonization has been identified as a risk factor for CRE colonization in ICU patients, as demonstrated in our current study [20]. Patients with acute myocardial infarction were admitted to isolated rooms in the cardiovascular ICU during the period following the intervention, thus potentially reducing their risk of CRE colonization. 

Probiotics exert their effects by producing short-chain fatty acids from metabolic precursors, leading to similar downstream effects, such as immune modulation and increased mucosal barrier function [18]. Moreover, they may have the added effect of producing their own antimicrobial compounds, physically occupying the epithelial niche, and limiting the ability of other pathogens to colonize the enteric microbiome; specifically, their main effect on the clearance or decolonization of MDRO is dysbiosis [7], which possesses the primary mechanism of CRE acquisition and expansion. Furthermore, patients with CRE showed low α-diversity, reduced anaerobic bacteria producing short-chain free acids, and increased *Enterobacteriaceae* [19]. Our previous study showed that patients with CRE exhibited dysbiosis associated with a higher relative abundance of *Enterobacteriaceae* [5]. Significantly, the reduction in dysbiosis through FMT can decrease the levels of CRE [6], and the effectiveness of FMT in clearing MDRO is influenced by the extent of dysbiosis [4,5]. Probiotics may serve as a therapeutic alternative for MDRO colonization by addressing dysbiosis, even in ICU patients [7,20]. 

Numerous clinical trials have shown the clearance effects of probiotics. However, many prior studies have not successfully demonstrated the preventive and clearing effects of MDRO [7,21,22]. Notably, three studies have shown the decolonizing effect of probiotics [23,24,25,26]. Regarding *Enterobacteriaceae*, only one randomized control trial showed eradication of ESBL *Enterobacteriaceae,* but the result was not statistically significant [27]. Furthermore, some nonrandomized studies have demonstrated clearance of CRE [9,28]. However, these studies did not encompass a large number of patients or an adequate follow-up period to effectively demonstrate the MDRO clearance efficacy of probiotics. Probiotics’ effectiveness might be less than that of FMT; hence, studies comparing them would benefit from larger sample sizes and longer durations. Yet, implementing such a methodology seemed impractical within the limits of our current study design. Furthermore, numerous risk factors for CRE colonization may have been impacted during the study period; thus, the studies were unable to conclusively establish the probiotics’ effectiveness in clearing or preventing CRE. In contrast, unlike others, our study showed the preventive effect of CRE through multivariable analysis, adjusting for additional factors, and reinforced our hypothesis with microbiome analysis.

However, the administration of probiotics to patients in the ICU remains a subject of debate. Nevertheless, a meta-analysis demonstrated that probiotics, when used in critically ill patients, decreased the length of ICU stay and the incidence of ventilator-associated pneumonia and antibiotic-induced diarrhea, including *C. difficile* colitis. This finding, though supported by low-quality studies, showed no impact on mortality [20,29]. Probiotics are believed to mitigate gut dysbiosis induced by broad-spectrum antibiotics, thereby inhibiting the proliferation of pathogenic bacteria in the ICU. This contributes to the prevention of nosocomial infections, sepsis, and organ failure. Additionally, they curb gut inflammation and the incursion of pathogens resulting from dysbiosis [20]. Moreover, some cases of fungemia have been reported, followed by *Saccharocmyces boulardii* administration. Nonetheless, no fungemia has been observed in ICU patients who were administered *Saccharocmyces boulardii* at our hospital [30]. For MDRO prevention, probiotic administration to patients was believed to be a reasonable option. 

Limited information is available regarding the comparative efficacy of *Saccharomyces boulardii* and *Lactobacillus rhamnosus* in MDRO decolonization. Research on rats demonstrated that neither of the probiotic strains, including *Lactobacillus rhamnosus* GG, *Saccharomyces boulardii*, nor *Pediococcus acidilacticii* C69, eradicated VRE colonization. However, they offered protection against VRE-associated epithelial damage, with *Saccharomyces boulardii* providing the most effective protection among the three probiotic microorganisms [31]. Another study focusing on Danish adults who traveled to India for 10–28 days revealed that *Lactobacillus rhamnosus* (Dicoflor®) did not significantly diminish the colonization rates of ESBL-Ent [8]. However, further research is necessary to ascertain the optimal probiotic regimen for the prevention and decolonization of MDRO colonization. A recent study demonstrated that *Saccharomyces boulardii* CNCM I-745 significantly diminished the incidence of travelers’ diarrhea [32]. Specifically, *Saccharomyces boulardii* exerts its preventive effect on antibiotic-associated diarrhea through three primary actions: the luminal action, the trophic action, and the anti-toxin effects [33,34]. The luminal action refers to the capacity to alter the gut environment by generating short-chain fatty acids, which may inhibit the growth of pathogenic bacteria. The trophic action pertains to the ability to encourage the proliferation of beneficial bacteria, such as *Lactobacillus* and *Bifidobacterium*, which may outcompete pathogenic bacteria for nutrients and space. The anti-toxin effects involve the capability to disrupt pathogenesis within the intestinal lumen by obstructing pathogen toxin receptor sites, serving as a decoy receptor for the pathogenic toxin, or directly neutralizing the pathogenic toxin.

Our study had several strengths. First, we demonstrated the risk of CRE colonization after adjusting for multiple risk factors during admission. Because our hospital has a CRE screening program every 15 days, new CRE colonizations were identified. Second, we revealed differences in the microbiome analysis of fecal samples from patients with CRE who were administered probiotics. Therefore, probiotics effectively inhibited dysbiosis as CRE colonization increased in the feces, thereby decreasing the abundance of CRE. We confirmed that the abundance of *Enterobacteriaceae* and *Klebsiella* decreased in patients treated with probiotics. 

However, this study had limitations. First, it was a retrospective study. Therefore, the results may have some bias and lower statistical power than those of prospective designs. However, we enrolled many patients and analyzed risk factors for CRE colonization using multivariate analysis. Second, there was a small number of patients with microbiome analysis, and there was no follow-up analysis after probiotics. Third, we could not identify the difference according to the probiotic type because of the number of *Lactobacillus rhamnosus*.

## 5. Conclusions

In conclusion, this study suggests that probiotics, such as *Saccharomyces boulardii or Lactobacillus rhamnosus*, may have a role in preventing new CRE colonization in the ICU, demonstrating the preventive effect through multivariable analysis and microbiome analysis. However, the administration of probiotics to ICU patients remains controversial, and further large randomized prospective studies are warranted to validate these findings. Finally, the study highlights the importance of managing risk factors to prevent CRE colonization in the ICU, emphasizing the need for a comprehensive approach to controlling CRE in healthcare settings.

## Figures and Tables

**Figure 1 microorganisms-11-02970-f001:**
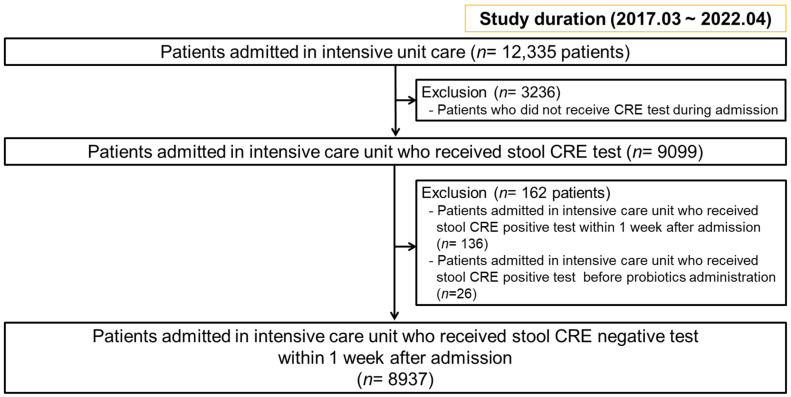
Flow chart of patients’ selection.

**Figure 2 microorganisms-11-02970-f002:**
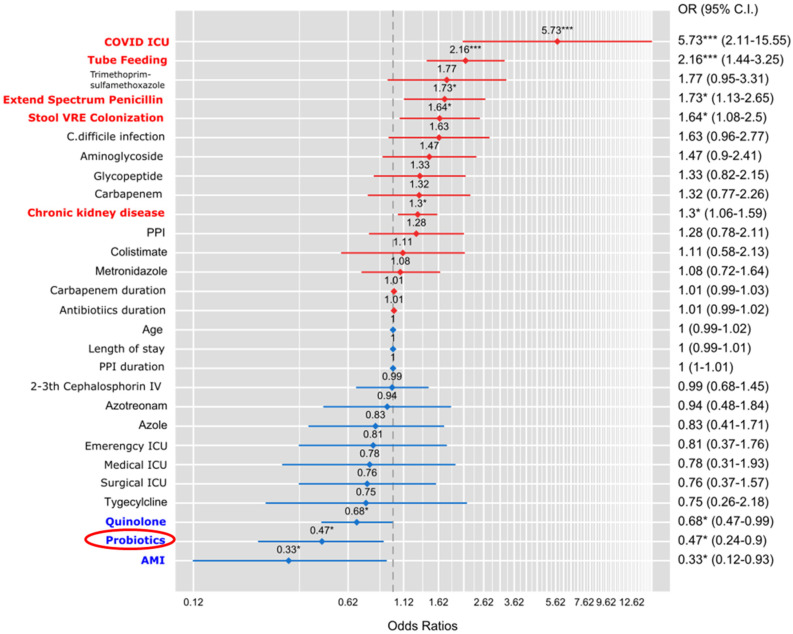
Multivariable analysis of the risk factors associated with carbapenem-resistant *Enterobacteriaceae* (CRE) colonization.; *, *p*-value < 0.05; and ***, *p*-value < 0.001; red color texts indicated the significant factors positively correlated with CRE colonization; blue color texts indicated the significant factors negatively correlated with CRE colonization; red oval emphasizes probiotics as a significant negative correlated factor.

**Figure 3 microorganisms-11-02970-f003:**
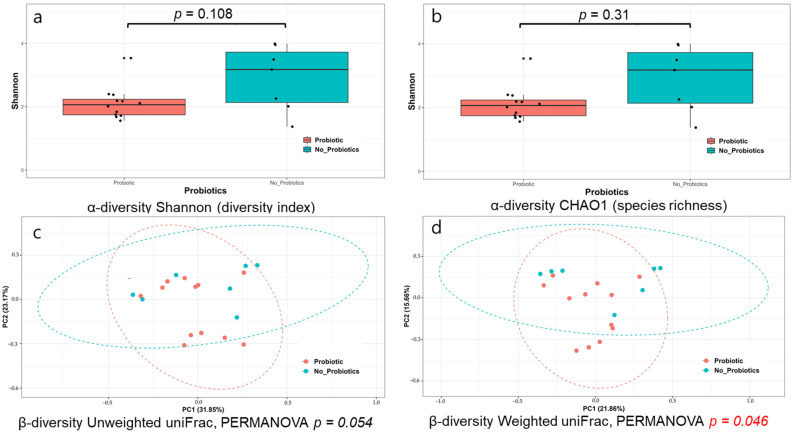
Microbiome analysis of 20 patients’ samples with CRE colonization, divided into probiotics (12 samples) and non-probiotics (8 samples) groups. (**a**) α-diversity (Shannon) between the probiotic and non-probiotic groups. (**b**) α-diversity (CHAO1) between the probiotic and non-probiotic groups. (**c**) β-diversity (unweighted uniFrac) between the probiotics group and non-probiotics group. (**d**) β-diversity (weighted uniFrac) between the probiotic and non-probiotic groups. The red color indicated a significant difference between the groups. Abbreviations: PERMANOVA, permutational multivariate analysis of variance.

**Figure 4 microorganisms-11-02970-f004:**
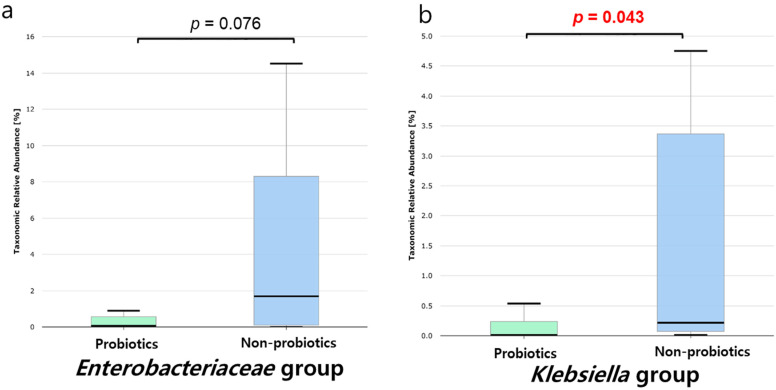
Microbiome analysis for patients with CRE colonization according to probiotics administration. (**a**) Relative abundance of the *Enterobacteriaceae* family between the probiotics and non-probiotics groups. (**b**) Relative abundance of the *Klebsiella* genus between the probiotic and non-probiotic groups. The red color indicated a significant difference between the groups.

**Table 1 microorganisms-11-02970-t001:** Baseline characteristics between carbapenem-resistant *Enterobacteriaceae* (CRE) colonization and no CRE colonization.

Characteristics	CRE Colonization (*n* = 157)	No CRE Colonization (*n* = 8780)	*p*-Value
Age, years, median (IQR)	73.0 (61.0–80.0)	69.0 (56.0–79.0)	0.006
Sex			>0.999
Male	91 (58.0)	5119(58.3)	
Female	73 (42.0)	3661 (41.7)	
ICU Category			<0.001
Cardiology ICU (CCU)	9 (5.2)	770 (8.8)	
Emergency ICU (EA and EB)	88 (56.1)	5778 (65.8)	
Medical ICU (MA and MB)	36 (22.9)	1491 (17.0)	
Surgical ICU (SB)	12 (7.6)	681 (7.8)	
COVID ICU (IICU)	12 (7.6)	58 (0.7)	
Length of stay, days, median (IQR)	33.0 (21.0–63.0)	12.0 (6.0–21.0)	<0.001
Stool VRE Positive	725 (8.3)	41 (26.1)	<0.001
*Clostridium difficile* infection	29 (18.5)	397 (4.5)	<0.001
CRE-colonized genus			
*Citrobacter*	3 (1.9)		
*Enterobacter*	9 (5.7)		
*Escherichia*	25 (15.9)		
*Klebsiella*	120 (76.3)		
Tube feeding	116 (66.7)	2558 (29.2)	<0.001
Probiotics	13 (8.3)	461 (5.3)	0.264
Probiotics duration, days, median (IQR)	0.0 (0.0–0.0)	0.0 (0.0–0.0)	0.165
*Saccharomyces boulardii*	10 (6.4)	340 (3.9)	0.455
*Lactobacillus rhamnosus*	3 (1.7)	121 (1.4)	0.955
PPI	135 (86.0)	6236 (71.0)	<0.001
PPI duration, days, median (IQR)	17.0 (2.0–42.0)	3.0 (0.0–10.0)	<0.001
Carbapenem	55 (35.0)	883 (10.1)	<0.001
Carbapenem duration, days, median (IQR)	14.0 (8.0–26.5)	9.0 (5.0–18.0)	<0.001
Antibiotics			
Aminoglycoside	32 (20.4)	548 (6.2)	<0.001
Penicillin	6 (3.8)	178 (2.0)	0.116
Extend Spectrum Penicillin	115 (73.2)	3675 (41.9)	<0.001
Macrolide	9 (5.7)	466 (5.3)	0.931
Glycopeptide	53 (33.8)	742 (8.5)	<0.001
Cephalosphorin PO	7 (4.5)	330 (3.8)	>0.999
1st Cephalosphorin IV	16 (10.2)	685 (7.8)	0.167
2nd–3rd Cephalosphorin IV	106 (67.5)	4662 (53.1)	<0.001
Quinolone	75 (47.8)	2967 (33.8)	<0.001
Metronidazole	75 (47.8)	2317 (26.4)	<0.001
Clindamycin	5 (3.2)	357 (4.1)	0.726
Azotreonam	16 (10.2)	264 (3.0)	<0.001
Colistimate	18 (11.5)	158 (1.8)	<0.001
Tygecylcline	8 (5.1)	76 (0.9)	<0.001
Trimethoprim–Sulfamethoxazole	20 (12.7)	181 (2.1)	<0.001
Rifamycin	6 (3.8)	169 (1.9)	0.248
Azole	13 (7.5)	193 (2.2)	<0.001
Antibiotics duration, days, median (IQR)	8 (1–16)	27 (17–50)	<0.001
Comorbidity			
Diabetes	4 (2.3)	351 (4.0)	0.344
Diabetes complication	1 (0.6)	73 (0.8)	0.484
Liver disease (Mild)	2 (1.3)	276 (3.1)	0.269
Liver disease (moderate to severe)	3 (1.9)	225 (2.6)	0.125
Acute myocardial infarction	4 (2.3)	908 (10.3)	0.002
Congestive heart failure	15 (9.6)	932 (10.6)	0.766
PAOD	6 (3.8)	229 (2.6)	0.490
Cerebral vascular attack	28 (17.8)	1842 (21.0)	0.389
Hemiplegia	22 (14.0)	1249 (14.2)	>0.999
Dementia	2 (1.3)	75 (0.9)	0.898
COPD	2 (1.3)	289 (3.3)	0.236
Rheumatic disease	3 (1.9)	46 (0.5)	0.074
Peptic ulcer	6 (3.8)	450 (5.1)	0.580
Chronic kidney disease	37 (23.6)	967 (11.0)	<0.001
Localized solid tumor	16 (10.2)	836 (9.5)	0.884
Lymphoma or Leukemia	2 (1.3)	60 (0.7)	0.690
Metastasis of tumor	2 (1.3)	243 (2.8)	0.374
HIV	0 (0.0)	7 (0.1)	>0.999
Charlson comorbidity index, median (IQR)	1 (0–2)	1 (0–2)	0.894

Data are presented as median (interquartile range) or number (). Abbreviations: IQR, interquartile range; COVID-19, coronavirus disease 2019; CRE, carbapenem-resistant *Enterobacteriaceae;* ICU, intensive care unit; VRE, vancomycin-resistance *enterococci;* PPI, proton pump inhibitor; HIV, human immunodeficiency virus; PAOD, peripheral arterial occlusive disease; and COPD, chronic obstructive pulmonary disease.

## Data Availability

The dataset presented in this study is available from the corresponding author upon reasonable request.

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
