# Peer review of "Role of Probiotics in Preventing Carbapenem-Resistant Enterobacteriaceae Colonization in the Intensive Care Unit: Risk Factors and Microbiome Analysis Study"

_microorganisms, 2023, doi:10.3390/microorganisms11122970_

Round 1

Reviewer 1 Report

Comments and Suggestions for Authors

The article presents an important topic with a solid and novel scientific approach. Below are some points to keep in mind to improve the manuscript.

If the study was retrospective, how did they access samples for metagenomics studies?
The statement that the samples come from another study and that it is on line 90 needs to be clarified.
How many samples were subjected to metagenomics studies? Are all of the patients included or only part of them? If they chose only a part, what was the selection criterion for the samples?
The total number of samples analyzed must be specified in Figure 3.
In line 194, they indicate a decrease in the relative abundance of the genus Escherichia; however, the statistical analysis showed that this difference was not significant.
Line 205 should be more precise about the measures to be incorporated in the ICU units that care for patients with COVID.

Author Response

Response to Reviewer 1’s comments

Comments to the Author

Comment 1-1:

If the study was retrospective, how did they access samples for metagenomics studies? The statement that the samples come from another study and that it is on line 90 needs to be clarified. How many samples were subjected to metagenomics studies? Are all of the patients included or only part of them? If they chose only a part, what was the selection criterion for the samples? The total number of samples analyzed must be specified in Figure 3.

Response: As we mentioned in this article, we obtained these samples from another study (KCT0004423) that tried to eliminate VRE or CRE colonization by fecal microbiota transplantation (FMT). Among the samples taken before receiving FMT, we divided the CRE stool samples according to whether probiotics were applied or not. Therefore, we can metagenomically evaluate the effect of probiotics on preventing CRE colonization.

Edited and added to the word is as follows:

“We conducted this study to investigate the FMT effect on decolonizing CRE or VRE [4]. From these samples, we selected the metagenomics data of 20 pre-FMT CRE stool samples. These samples were further categorized based on the application of probiotics into two groups: probiotic (12 samples) and non-probiotic (8 samples).” (line 94-97)

And edited to the words is as follows:

Figure 3 Microbiome analysis for 20 patients’ sample with CRE colonization dividing into probiotics (12 samples) and non-probiotics (8 samples) group.” (line 195-196)

Comment 1-2:

In line 194, they indicate a decrease in the relative abundance of the genus Escherichia; however, the statistical analysis showed that this difference was not significant.

Response: W When we reviewed the samples that underwent metagenomic analysis, we found that the main cause of the CRE genus was Klebsiella, not Escherichia. Therefore, the main causative genus might affect the difference according to probiotic administration. Additionally, when analyzing the CRE genus composition, Klebsiella (120, 76.3%) was the largest portion of CRE (Table 1). We also analyzed other genera in the Enterobacteriaceae family, such as Citrobacter and Enterobacter, and presented these results in Supplementary Figure 3. The differences in these genera were not significant, which might be due to their very relative abundance in both groups.

Edited to the word is as follows:

“Further, we analyzed the relative abundances of both groups, including the Enterobacteriaceae family (Figure 4a) and the Klebsiella genus (Figure 4b), which are the main taxa of CRE. The relative abundances of the Enterobacteriaceae family in the probiotics group were lower than those in the non-probiotics group (P = 0.076). Similarly, the relative abundance of the Klebsiella genus in the probiotics group was significantly lower than that in the non-probiotics group (P = 0.043). However, the relative abundances of the Escherichia, Citrobacter, and Enterobacter genera were not significantly different (P=0.673, P=0.316, and P=0.190, respectively) (Supplementary Figure 3)” (line 192-199) 

“Although we cannot directly demonstrate a decrease in the amount of CRE, the relative abundance of the Enterobacteriaceae family in patients who received probiotics showed a reduction in the number of these taxa. In particular, the abundance of the Klebsiella genus, the main genus in CRE, significantly decreased” (line 213-217)

Added to the manuscript is supplementary Figure 3

Comment 1-3:

Line 205 should be more precise about the measures to be incorporated in the ICU units that care for patients with COVID.

Response: As your detailed comments, we suggested various CRE prevention measures in the ICU patients, including COVID-19 patients with reference. 

Edited and added to the word is as follows:

“Various well-established prevention strategies can be implemented, such as hand hygiene, contact precautions, device control, environmental cleaning, and the early detection of CRE carriers using active surveillance methods [2]. In our hospital, UV light decontamination was routinely performed in the COVID-19 ICU to prevent the spread of CRE [19].” (line 232-236)

Reviewer 2 Report

Comments and Suggestions for Authors

Dear Author 

Good article and nice work. However, many things need to be added. 

1. It is not clear whether or not the participant has been started on antibiotics medication, and if yes, duration!! (is your participant under antibiotics for 3 to 4 days? Most of the gut bacteria will be gone)

2. Microbiome analysis is not completed as you showed only Alpha and Beta diversity  (you need to add composition analysis)

3. I am not sure how COVID-19 impacted your study because your data collected from March 2017 and April 2022 

4. you need to analyse your microbiome data differently by using risk factors

5. In your material and methods, it is not clear if you used 16s rRNA or whole genome sequence 

6. your microbiome analysis is better for analysing the species level of bacteria

7. is there any effect of other antibiotic-resistant bacteria on your study?

8. more relevant articles need to be added, especially if any related to ICU and Bacterial resistance 

9. which version of QIme was used? If not version 2, I would like to see the result analysed by Version 2. 

Comments on the Quality of English Language

Well written  Moderate editing of the English language required

Author Response

Response to Reviewer 2’s comments

Comments to the Author

Comment 2-1:

  1. It is not clear whether or not the participant has been started on antibiotics medication, and if yes, duration!! (is your participant under antibiotics for 3 to 4 days? Most of the gut bacteria will be gone)

Response: As you recommended, we additionally analyzed the duration of antibiotics and inserted the result into Table 1, which shows the baseline characteristics between CRE colonization and no CRE colonization. We also performed logistic analysis by adding the duration of antibiotics and presented the results in Figure 2. The results were marginally significant in multivariable analysis.

Edited to the manuscript are as follows:

“Among antibiotics, the frequencies of aminoglycosides, extended-spectrum penicillin, glycopeptides, second- and third-generation cephalosporins, quinolone, metronidazole, aztreonam, colistimate, tigecycline, trimethoprim-sulfamethoxazole, azoles, and anti-biotics duration were significantly different between the groups.” (lines 144-148)

Comment 2-2:

  1. Microbiome analysis is not completed as you showed only Alpha and Beta diversity (you need to add composition analysis)

Response: As you recommended, we added the comparison of family and genus in CRE patients between probiotics groups and non-probiotics groups (supplementary Fig. 1).

Added to the manuscript are as follows:

“We analyzed the family and genus composition of CRE-colonized stool according to probiotics administration (Supplementary Figure 1). The abundance of the Enterobacteriaceae family, to which CRE belongs, was lower in the probiotics group (3.43%) than in the non-probiotics group (20.85%). In contrast to the CRE genus composition, the abundances of Klebsiella genus (<1 vs. 6.15%), Escherichia (2.51 vs. 1.76 %), and Citrobacter (<1 vs. 4.04 %) were different between the probiotics group and non-probiotics group.” (line 171-176)

And added supplementary Figure 1

Comment 2-3:

  1. I am not sure how COVID-19 impacted your study because your data collected from March 2017 and April 2022

Response: According to our study, the COVID ICU was the most significant risk factor for CRE colonization. Longer ICU stays, carbapenem treatment, corticosteroid treatment, and invasive mechanical ventilation were commonly applied to COVID-19 patients, making them vulnerable to CRE acquisition. Additionally, many elderly patients admitted to the COVID ICU were possibly initially undetectable CRE carriers from long-term care facilities in the Korean medical environment. Infection prevention management in the COVID-ICU was also very difficult due to the need for protective clothing and the risk of COVID-19 infection. Since our study period included the COVID-19 pandemic, COVID-19 related factors might have impacted the results of our study.

Edited and added to the manuscript are as follows:

“Prolonged hospitalization, broad-spectrum antibiotics, immunosuppressed status, shared environments, and close proximity were known as CRE acquisition risk factors [15,16]. Longer lengths of ICU stays, carbapenem treatment, corticosteroid treatment, and invasive mechanical ventilation were frequently applied to COVID-19 ICU patients, increasing their susceptibility to CRE acquisition [17]. Additionally, elderly patients admitted to the COVID ICU were likely to be initially undetectable CRE carriers from long-term care facilities in the Korean medical environment [18]. Infection prevention management in the COVID ICU was also challenging due to the wearing of protective clothing and the risk of COVID-19 infection. Therefore, we should be more careful of CRE colonization in COVID-19 patients in ICU. Various well-established prevention strategies can be implemented, such as hand hygiene, contact precautions, device control, environmental cleaning, and the early detection of CRE carriers using active surveillance methods [2]. "(line 223-235)

Comment 2-4:

  1. you need to analyze your microbiome data differently by using risk factors

Response: Unfortunately, we only obtained microbiome data from a few patients in our previous study. Therefore, it is difficult to analyze it as a risk factor. Instead, the initial CRE colonization genus was added to Table 1.

Added to the manuscript are as follows:

“Analyzing the genus composition of CRE colonization, the Klebsiella genus (120, 76.3%) constituted the largest portion of CRE.” (lines 154-155)

Comment 2-5:

  1. In your material and methods, it is not clear if you used 16s rRNA or whole genome sequence

Response: As per your comment, we prescribed the following text in detail to reveal that we use 16S rRNA amplicon sequencing with v3-4 primers.

Edited to the manuscript are as follows:

“16S rRNA amplicon sequencing with v3-4 primers was performed using a MiSeq sequencer on an Illumina platform (Macrogen Inc., South Korea) according to the manufacturer's specifications.” (lines 106-108)

Comment 2-6:

  1. your microbiome analysis is better for analysing the species level of bacteria

Response: As far as we know, analyzing the species level is difficult with 16S rRNA V3-V4 amplicon sequencing with QIIME2. Therefore, we reanalyzed the data using the EzBioCloud tool with PKSSU 4.0 database, which some researchers have used to analyze the species level. The species composition was added to our current study using an analysis program (Supplementary Figure 2).

Edited to the manuscript are as follows:

“QIIME 2 and EzBioClould (CJ Bioscience Inc, Seoul, Korea; https://www.ezbiocloud.net/contents/16smtp) were used for OTU analysis and taxonomy [11,12]. The major sequence of each OTU was identified, and high-quality sequence reads were assigned to “species group” at 97% sequence similarity using the PKSSU4.0 database.   “ (lines 122-125)

Added to the manuscript is supplementary Figure 2. “Comparison of species abundance in carbapenem-resistant Enterobacteriaceae colonized patients according to probiotics administration.”

Comment 2-7:

  1. is there any effect of other antibiotic-resistant bacteria on your study?

Response: As per your comment, representative antibiotic-resistant bacteria were analyzed (Table 1) and included in the multivariate analysis (Fig. 3), including stool VRE colonization and Clostridium difficile infection. Stool VRE colonization was an independent risk factor for CRE acquisition. Glycopeptide and length of stay were no longer significant risk factors after performing multivariable analysis again. Instead, AMI and quinolone were newly found to correlate with CRE colonization negatively. Therefore, we will discuss these factors.

Edited to the manuscript are as follows:

“. In multivariate analysis, Coronavirus Disease 2019 ICU, tube feeding, antibiotics such as extended-spectrum penicillin, stool vancomycin-resistance enterococci colonization, and chronic kidney disease were significantly associated with de novo CRE infection” (line 22-25)

“Age, ICU category, total length of stay, stool vancomycin-resistance enterococci (VRE) colonization, Clostridium difficile infection, proton pump inhibitor usage and duration, carbapenem usage and duration, and tube feeding proportion significantly differed between the two groups.” (line 165-168)

“In multivariate analysis, Coronavirus Disease 2019 (COVID-19) ICU (odds ratio [OR], 5.73; 95% confidence interval [CI], 2.11–15.55), tube feeding (OR, 2.16; 95% Cl: 1.44–3.25),), ex-tend spectrum penicillin (OR, 1.73; 95% Cl: 1.13–2.65), stool VRE colonization (OR, 1.64; 95% Cl: 1.08–2.5), and chronic kidney disease (OR, 1.3; 95% Cl: 1.06–1.59) were positively correlated with new CRE colonization. Moreover, quinolone (OR, 0.68; 95% Cl: 0.47–0.99), acute myocardial infarction (OR, 0.33; 95% Cl: 0.12–0.93), and probiotics administration (OR, 0.47; 95% Cl: 0.24–0.9) significantly and negatively correlated with new CRE colonization..” (line 159-166)

“Additionally, stool VRE colonization has been identified as a risk factor for CRE colonization in ICU patients, as demonstrated in our current study [20]. Patients with acute myocardial infarction were admitted to isolated rooms in the cardiovascular ICU during the period following the intervention, thus potentially reducing their risk of CRE colonization.” (line 236-240)

Comment 2-8:

  1. more relevant articles need to be added, especially if any related to ICU and Bacterial resistance

Response: As you recommended, we discussed the preventive strategy for CRE spreading with some references.

Added to the manuscript are as follows:

“Various well-established prevention strategies can be implemented, such as hand hygiene, contact precautions, device control, environmental cleaning, and the early detection of CRE carriers using active surveillance methods [2]. In our hospital, UV light decontamination was routinely performed in the COVID-19 ICU to prevent the spread of CRE [19]. Additionally, stool VRE colonization has been identified as a risk factor for CRE colonization in ICU patients, as demonstrated in our current study [20].”

“19.       Lowman, W.; Etheredge, H.R.; Gaylard, P.; Fabian, J. The novel application and effect of an ultraviolet light decontamination strategy on the healthcare acquisition of carbapenem-resistant Enterobacterales in a hospital setting. J Hosp Infect 2022, 121, 57-64, doi:10.1016/j.jhin.2021.12.008. (

  1. Papadimitriou-Olivgeris, M.; Spiliopoulou, I.; Christofidou, M.; Logothetis, D.; Manolopoulou, P.; Dodou, V.; Fligou, F.; Marangos, M.; Anastassiou, E.D. Co-colonization by multidrug-resistant bacteria in two Greek intensive care units. Eur J Clin Microbiol Infect Dis 2015, 34, 1947-1955, doi:10.1007/s10096-015-2436-4.” (line 376-381)

Comment 2-9:

  1. which version of QIme was used? If not version 2, I would like to see the result analysed by Version 2.

Edited to the manuscript are as follows:

Response: As we mentioned before, we indeed used QIMME 2 and Ezbiocloud for taxonomic analysis for the species level.

“QIIME 2 and EzBioClould (CJ Bioscience Inc, Seoul, Korea; https://www.ezbiocloud.net/contents/16smtp) were used for OTU analysis and taxonomy [11,12]. The major sequence of each OTU was identified, and high-quality sequence reads were assigned to “species group” at 97% sequence similarity using the PKSSU4.0 data-base.“ (line 122-125)

Reviewer 3 Report

Comments and Suggestions for Authors

On request of Microorganisms, I have revised the manuscript titled “Analysis of Risk factors for Carbapenem-Resistant Enterobacteriaceae Colonization in the Intensive Care Unit and Study of the Effect of Probiotics to Prevent Colonization”, by Jung-Hwan Lee and co-workers.

In this single-centred retrospective study, the authors have first analysed the risk factors for infections sustained by carbapenem-resistant Enterobacteriaceae (CRE) in the intensive care unit (ICU), where older patients with multiple comorbidities are admitted. Secondly, the preventive effect of probiotics administration for colonization has been verified.

GENERAL COMMENTS

To limit the emerging problem of the worrying increase in the occurrence of multidrug-resistant organisms, including carbapenem-resistant Enterobacteriaceae (CRE), especially in the intensive care units, where older and immunocompromised patients are hospitalized is of paramount importance. In this context, studies finalized to the evaluation of factors that could promote or prevent colonization by such bacteria are welcome.

Some minor issues should be addressed before publication.

Line 28. Please, check the font size.

Line 60. Please, specify MDRO.

Figure 1. Please, add the zero before 4 in the dates of the study.

Line 98. Please, correct Escheria with Escherichia.

Figure 2. There is no specification for *, and ***, while ** is missing. I do not understand. Please, clarify.

Line 165. Please, correct Fig. with Figure, and check all manuscript to correct similar issues if present.

Figure 4, panel (a). Please, correct Entrobacteriaceae with Enterobacteriaceae.

Please, remove spaces between the references’ numbers.

Both in the text and in the Figures captions, use the Greek letters in place of “alfa” or “beta”.

Line 239. Please, insert a space between “C.” and “difficile”.

Line 255. Please, correct Eschechia with Escherichia.

Conclusions are very poor and need improvement.

All references should be rewritten to respect the template of Microorganisms.

Comments on the Quality of English Language

 Minor editing of English language required.

Author Response

Response to Reviewer 3’s comments

Comments to the Author

Comment 3-1:

Line 28. Please, check the font size.

Response: As you commented, we collected the font style and size. 

Comment 3-2:

Line 60. Please, specify MDRO.

Response: As your comment, we specify MDRO

Edited to the manuscript are as follows:

“,, multi-drug resistant organism (MDRO) colonization.” (line 60)

Comment 3-3:

Figure 1. Please, add the zero before 4 in the dates of the study.

Response: As your comment, we added the zero before 4 on the study date.

Edited to the figure are as follows:

Comment 3-4:

Line 98. Please, correct Escheria with Escherichia.

Response: As your comment, we collected Escheria with Escherichia.

Edited to the manuscript are as follows:

“…such as the Enterobacteriaceae family and the genera Klebsiella, Escherichia, Citrobacter, and Enterobacter” (lines 103-104)

Comment 3-5:

Figure 2. There is no specification for *, and ***, while ** is missing. I do not understand. Please, clarify.

Response: As your comment, we clarified *, **, and ***.

Edited to the manuscript are as follows:

*, P-value <0.05 ; **, P-value <0.01;***, P-value <0.001” (line 169)

Comment 3-6:

Line 165. Please, correct Fig. with Figure, and check all manuscript to correct similar issues if present.

Response: As your comment, we collect Fig. with Figure by checking all manuscripts.

Comment 3-7:

Figure 4, panel (a). Please, correct Entrobacteriaceae with Enterobacteriaceae.

Response: As your comment, we collect Entrobacteriaceae with Enterobacteriaceae.

Comment 3-8:

Please, remove spaces between the references’ numbers.

Response: As your comment, we remove spaces between the references’ numbers

Comment 3-9:

Both in the text and in the Figures captions, use the Greek letters in place of “alfa” or “beta”.

Response: As your comment, we use the Greek letters in place of “alfa” or “beta” in both the text and the Figures captions,

Comment 3-10:

Line 239. Please, insert a space between “C.” and “difficile”.

Response: As your comment, we insert a space between “C.” and “difficile”.

Comment 3-11:

Line 255. Please, correct Eschechia with Escherichia.

Response: We confirmed that the relative abundance of Escherichia did not decrease so that we remove this word in the manuscript

Edited to the manuscript are as follows:

“In particular, the abundance of the Klebsiella genus, the main genus in CRE, significantly decreased.” (line 216-217)

Comment 3-12:

Conclusions are very poor and need improvement.

Response: We re-write the conclusion of this article focusing on the perspective of the CRE prevention effect of probiotics and the need to control CRE risk factors in the ICU.

Edited to the manuscript are as follows:

“In conclusion, this study suggests that probiotics, such as Saccharomyces boulardii or Lactobacillus rhamnosus, may have a role in preventing new CRE colonization in the ICU, demonstrating the preventive effect through multivariable analysis and microbiome analysis. However, the administration of probiotics to ICU patients remains controversial, and further large randomized prospective studies are warranted to validate these findings. The study also highlights the importance of managing risk factors to prevent CRE colonization in the ICU, emphasizing the need for a comprehensive approach to controlling CRE in healthcare settings.” (line 299-306)  

Comment 3-13:

All references should be rewritten to respect the template of Microorganisms..

Response: As your comment, we collected reference formats according to Microorganisms.

Round 2

Reviewer 2 Report

Comments and Suggestions for Authors

Moderate editing of English language required

Comments on the Quality of English Language

Moderate editing of English language required

Author Response

Thank you for your careful recommendation. 

As your comment, we sent our article to the English editing service and received a collected one. We attached the certification of editing.
